# Advantages of Multiposition Scanning in Echocardiographic Assessment of the Severity of Discordant Aortic Stenosis

**Elena Zelikovna Golukhova** [1], **Inessa Viktorovna Slivneva** [2,*], **Inga Yur'evna Farulova** [3], **Ivan Ivanovich Skopin** [3], **Damir Ildarovich Marapov** [4], **Dar'ya Vladimirovna Murysova** [3], **Yuliya Dmitrievna Pirushkina** [5] **and Irina Vasilyevna Volkovskaya** [6]

[1] A.N. Bakulev National Medical Scientific Center for Cardiovascular Surgery, Ministry of Health of the Russian Federation, 121552 Moscow, Russia; ezgolukhova@bakulev.ru

[2] Department of Cardiovascular and Comorbid Pathology, A.N. Bakulev National Medical Scientific Center for Cardiovascular Surgery, Ministry of Health of the Russian Federation, 121552 Moscow, Russia

[3] Department of Reconstructive Surgery of Heart Valves and Coronary Arteries, A.N. Bakulev National Medical Research Center for Cardiovascular Surgery, Ministry of Health of the Russian Federation, 121552 Moscow, Russia; iyfarulova@bakulev.ru (I.Y.F.); iiskopin@bakulev.ru (I.I.S.); dvmurysova@bakulev.ru (D.V.M.)

[4] Department of Public Health, Economics and Health Care Management, Kazan State Medical Academy—Branch Campus of the Federal State Budgetary Educational Institution of Further Professional Education, Russian Medical Academy of Continuous Professional Education, Ministry of Healthcare of the Russian Federation, 420012 Kazan, Russia; damirov@list.ru

[5] Department of Cardiology and Functional Diagnostics, A.N. Bakulev National Medical Scientific Center for Cardiovascular Surgery, Ministry of Health of the Russian Federation, 121552 Moscow, Russia; pirushkina4444@mail.ru

[6] Polyclinic Department of the Institute of Coronary and Vascular Surgery, A.N. Bakulev National Medical Scientific Center for Cardiovascular Surgery, Ministry of Health of the Russian Federation, 121552 Moscow, Russia; iivolkovskaya@bakulev.ru

* Correspondence: ivslivneva@bakulev.ru

**Abstract:** Aim of the study: The aim of this study was to perform a comparative analysis of severity of discordant aortic stenosis (AS) assessment using multiposition scanning and the standard apical window. Materials and Methods: All patients ($n$ = 104) underwent preoperative transthoracic echocardiography (TTE) and were ranked according to the degree of AS severity. The reproducibility feasibility of the right parasternal window (RPW) was 75.0% ($n$ = 78). The mean age of the patients was 64 years, and 40 (51.3%) were female. In 25 cases, low gradients were identified from the apical window not corresponding to the visual structural changes in the aortic valve, or disagreement between the velocity and calculated parameters was detected. Patients were divided into two groups: concordant AS ($n$ = 56; 71.8%) and discordant AS ($n$ = 22; 28.2%). Three individuals were excluded from the discordant AS group due to the presence of moderate stenosis. Results: Based on the comparative analysis of transvalvular flow velocities obtained from multiposition scanning, the concordance group showed agreement between the velocity and calculated parameters. We observed an increase in the mean transvalvular pressure gradient ($\Delta P_{mean}$) and peak aortic jet velocity ($V_{max}$), $\Delta P_{mean}$ in 95.5% of patients, velocity time integral of transvalvular flow (VTI AV) in 90.9% of patients, and a decrease in aortic valve area (AVA) and indexed AVA in 90.9% of patients after applying RPW in all patients with discordant AS. The use of RPW allowed the reclassification of AS severity from discordant to concordant high-gradient AS in 88% of low-gradient AS cases. Conclusion: Underestimation of flow velocity and overestimation of AVA using the apical window may lead to misclassification of AS. The use of RPW helps to match the degree of AS severity with the velocity characteristics and reduce the number of low-gradient AS cases.

**Keywords:** TTE; transthoracic echocardiography; discordant aortic stenosis; right parasternal window; multiple-view scanning of aortic valve; aortic root angulation; reclassification of aortic stenosis

## 1. Introduction

Severe aortic stenosis (AS) is a condition characterized by a mean transvalvular pressure gradient ($\Delta P_{mean}$) of $\geq$40 mmHg, peak aortic jet velocity ($V_{max}$) of $\geq$4 m/s, aortic valve area (AVA) of $\leq$1.0 cm$^2$, and indexed aortic valve area (AVAi) of $\leq$0.6 cm$^2$/m$^2$ [1–3]. When these parameters are in agreement, they are referred to as concordant, indicating that there is an alignment of velocity characteristics with structural alterations of the valve or with the effective orifice area (EOA). However, about 40% of AS patients exhibit discordant AS, where Doppler echocardiography or other types of measurements yield conflicting results. Typically, this is associated with low-gradient AS, and can complicate the assessment of the severity of stenosis and make it difficult to determine an appropriate treatment strategy [4].

The apical window is commonly utilized for measuring transaortic flow velocity and pressure gradients. However, inaccurate measurements of velocity characteristics and the residual orifice area may result in misclassification of AS severity, and consequently lead to inappropriate management of patients [5]. Thus, this study aimed to conduct a comparative analysis of the multiple-view transthoracic echocardiography (TTE) for assessing discordant AS severity, and to reassess the severity of AS by incorporating the right parasternal view.

## 2. Materials and Methods

The study was a prospective, single-center, observational (cohort) study aimed at assessing the severity of AS using TTE. The study protocol was approved by the Local Ethics Committee. Patients were included in the study if they were over 18 years of age and had organic aortic valve (AV) lesions with Echo signs of moderate or severe stenosis. Patients were excluded from the study if they had subvalvular or supravalvular obstruction, inadequate visualization, severe chest deformity, active infective endocarditis, or previous "open" heart surgery.

The initial cohort of the study consisted of 104 patients with moderate, severe, or very severe AS. The reproducibility feasibility of the right parasternal window (RPW) was determined to be 75.0% (*n* = 78). Table 1 presents the clinical characteristics of the patients before surgery. The mean age of the patients was 64 [50; 70] years, and there were no significant gender differences. Arterial hypertension was the most prevalent comorbid condition (*n* = 67.9%). The proportion of atherosclerotic lesions in cerebral and peripheral arteries was similar (*n* = 24.3%). Significant coronary pathology was present in 10.3% of cases.

**Table 1.** Preoperative characteristics of patients.

| Parameters | | Baseline | Min | Max |
|---|---|---|---|---|
| Age, years | | 64 [55; 70] | 20 | 81 |
| Gender: | -Male | 38 (48.7) | | |
| | -Female | 40 (51.3) | | |
| BSA (m$^2$) | | 1.94 [1.81; 2.07] | 1.49 | 2.72 |
| BMI (kg/m$^2$) | | 28.1 [24.6; 31.2] | 16.9 | 45.3 |
| Rhythm: | -Sinus | 73 (93.6) | | |
| | -Paroxysmal atrial fibrillation | 1 (1.3) | | |
| | -Persistent atrial fibrillation | 4 (5.1) | | |
| Concomitant pathology | | | | |
| Arterial hypertension | I grade | 4 (5.1) | | |
| | II grade | 8 (10.3) | | |
| | III grade | 41 (52.6) | | |
| Atherosclerotic disease of great vessels | | 19 (24.3) | | |
| Atherosclerotic disease of peripheral vessels | | 19 (24.3) | | |
| COPD | | 10 (12.8) | | |
| Bronchial asthma | | 1 (1.3) | | |
| Diabetes mellitus | | 8 (10.3) | | |

**Table 1.** *Cont.*

| Parameters | | Baseline | Min | Max |
|---|---|---|---|---|
| Chronic kidney disease | | 4 (5.1) | | |
| History of cerebral stroke/TIA | | 1 (1.3) | | |
| Coronary artery disease (stenosis ≥ 65%) | | 8 (10.3) | | |
| History of myocardial infarction | | 4 (5.1) | | |
| | NYHA II | 13 (16.7) | | |
| Functional class | NYHA III | 63 (46.2) | | |
| | NYHA IV | 2 (2.6) | | |
| EuroScore II, (%) | | 1 [1; 2] | 1 | 7 |

BSA—body surface area, BMI—body mass index, COPD—chronic obstructive pulmonary disease, TIA—transient ischemic attack, NYHA—New York Heart Association. Data are presented as absolute values (*n*) and percentages (%), median (Me), and interquartile ranges [IQR].

## 2.1. Echocardiography Analysis

Transthoracic echocardiography was performed using a PHILIPS EPIQ CVx cardiac ultrasound system with an X5-1 transducer. Preoperative echocardiography was performed by two cardiovascular imaging specialists.

Quantitative measurements and assessment of left ventricle (LV) contractile function (biplane Simpson method) were performed according to the 2015 guidelines of the American Society of Echocardiography and the European Association of Cardiovascular Imaging (ASE and EACVI) [6].

The following measures, obtained by continuous-wave Doppler, were assessed: $V_{max}$, mean pressure gradient ($\Delta P_{mean}$) [1]. AS area was calculated as $AVA(EOA) = \frac{CSA_{LVOT} \times VTI_{LVOT}}{VTI_{AV}}$ (Figure 1), where $CSA_{LVOT}$ is the cross-sectional area of the left ventricle outflow tract, $VTI_{LVOT}$ is the left ventricle outflow tract velocity time integral, and $VTI_{AV}$ is the velocity time integral of transvalvular flow. The LVOT diameter was measured at the same distance (0.5–1.0 cm) from the AV as the control volume position of the pulsed-wave Doppler. The AVAi was then calculated.

Patients were ranked according to the severity of AS, following the recommendations of the EACVI and the ASE from 2017 [1]. In case of discordant values of $V_{max}$ and $P_{mean}$, the severity of AS was determined by the higher parameter. The estimation of EOA by the continuity equation depended on the variability of measurements, including the variability of data during recording; therefore, AVA and AVAi were considered as auxiliary criteria for ranking.

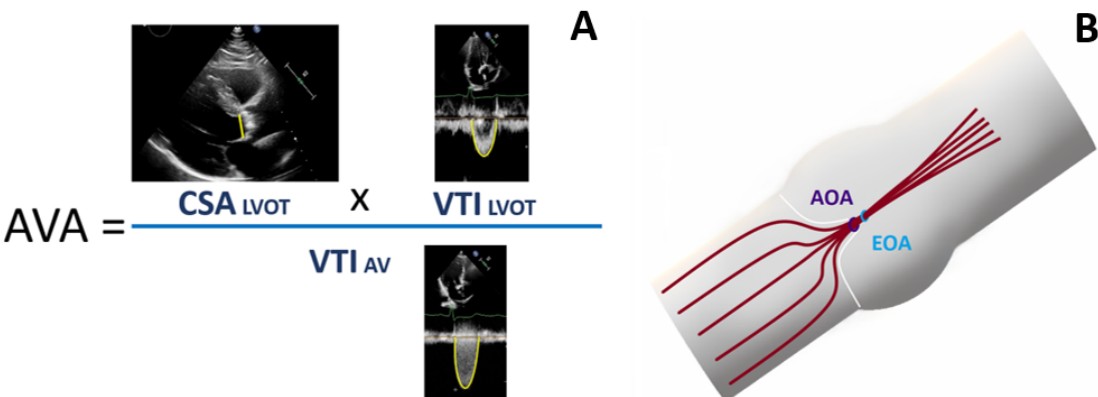

**Figure 1.** Equation of flow continuity. (**A**) Calculation of aortic valve area, (**B**) the effective area (EOA) is a hemodynamic parameter of the aortic stenosis severity, and in the majority of cases, EOA is smaller than the anatomical orifice (AOA).

The study evaluated the aortic root angulation in the parasternal long-axis view of the LV. To achieve this, the angle between the median plane of the aortic root and the plane of the interventricular septum was measured (as shown in Figure 2A). In addition,

the interventricular septal thickness at the basal level and Doppler intercept angle were measured using the apical 5-chamber view (A5C) (as shown in Figure 2B).

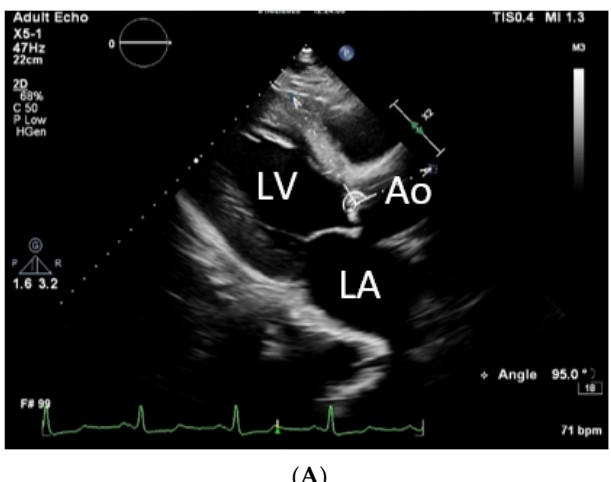

(**A**)

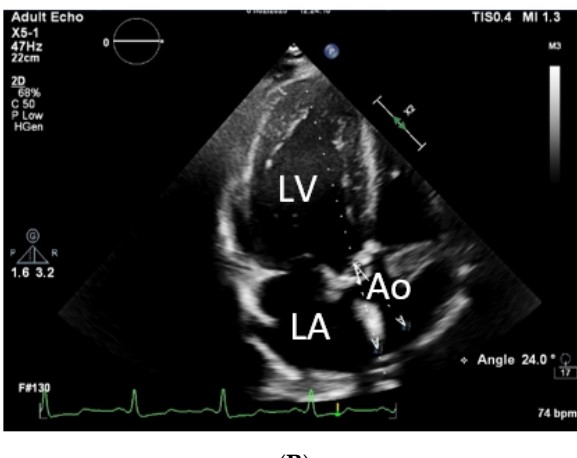

(**B**)

**Figure 2.** (**A**) Left ventricular long-axis view, aortic root angulation (the aortoseptal angle is 95°); (**B**) apical 5-chamber view (the Doppler transaortic flow intercept angle is 24°). The left ventricle (LV), left atrium (LA), and ascending aorta (Ao) are registered.

## 2.2. Multiple-View Scanning of the Aortic Valve

The echocardiographic examination was conducted with all patients initially placed in a left lateral position, with the left arm raised and bent at the elbow. Following this, the patient was repositioned onto their right side. The right parasternal scanning window was usually positioned 1–2 intercostal spaces higher than the left parasternal window (Figure 3). In some cases, additional rotation of the patient to the right was required to optimize the image. The ascending aorta and AV were detected using the RPW in order to provide an optimal Doppler readout of the oncoming transvalvular flow (Figure 4).

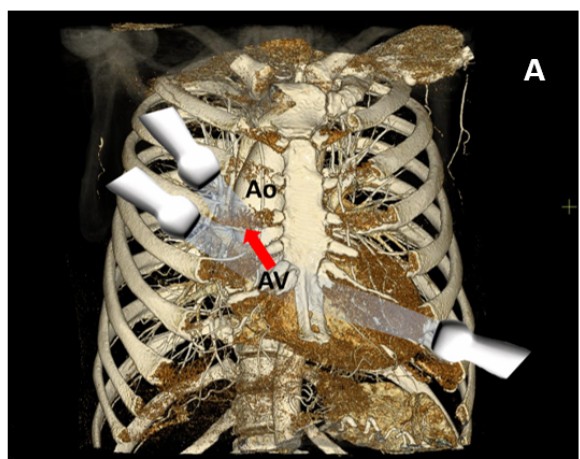

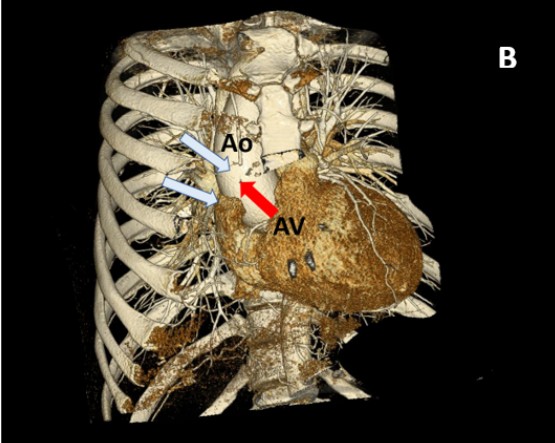

**Figure 3.** Doppler transaortic flow intercept. (**A**) Apical and right-parasternal positioning of the sector ultrasound transducer, (**B**) facing direction of the ultrasound beam plane in relation to the transaortic flow (marked by a red arrow). The aortic valve (AV) and ascending aorta (Ao) are registered.

Five consecutive rhythm-averaged complexes, excluding post-extrasystolic potentiation, were evaluated in the presence of arrhythmia [1].

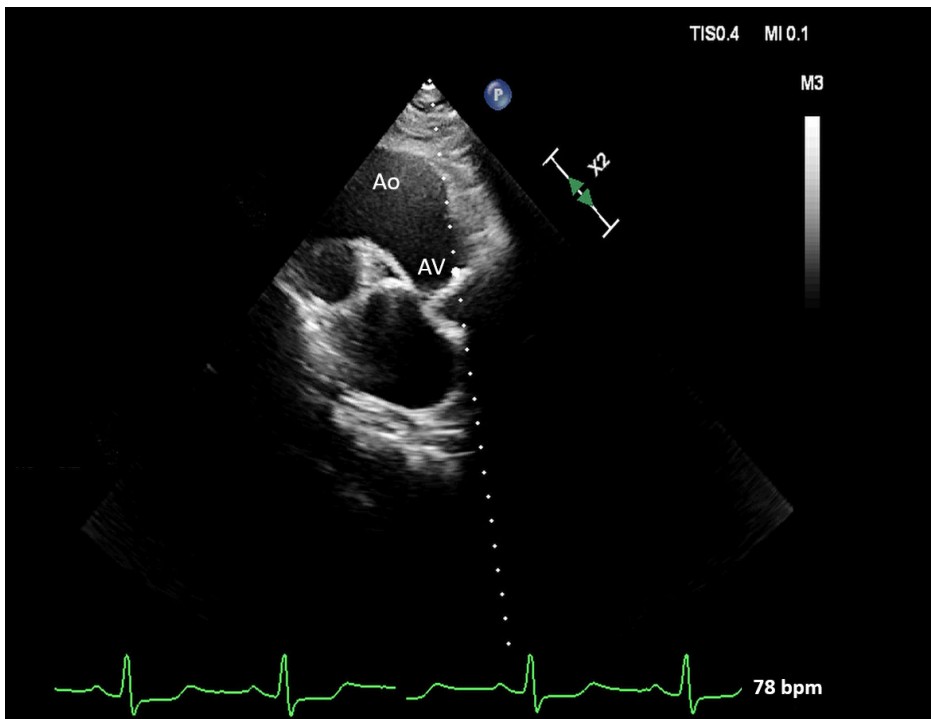

**Figure 4.** Right parasternal view provides an optimal Doppler readout of the counter transaortic flow. The ascending aorta (Ao) and aortic valve (AV) are registered.

### 2.3. Reproducibility

Two echocardiographers independently estimated pre-selected standard echo images of 10 random patients. The images were re-measured by the first (intra-observer variability) and by the second (interobserver variability) observer. Test–retest reliability was evaluated 2 weeks after the first analysis. Intra- and interobserver variability, as well as test–retest reliability, of various parameters in the selected images were calculated using the intraclass correlation coefficient (ICC).

### 2.4. Statistical Analysis

Quantitative variable distribution normality was assessed using the Shapiro–Wilk test for sample sizes less than 50 or the Kolmogorov–Smirnov test for sample sizes greater than 50. For non-normally distributed data, the median (Me) and lower and upper quartiles (Q1; Q3) were reported. Categorical data are presented as absolute values and percentages. The Mann–Whitney U-test was employed to compare two groups on a quantitative variable with a non-normal distribution, while the Wilcoxon test was used to compare linked samples with non-normal quantitative parameters. Statistical analysis was conducted using IBM SPSS Statistics v. 26 (IBM Corporation, Armonk, NY, USA) and StatTech v. 3.0.9 (StatTech LLC, Kazan, Russia).

## 3. Results

The patients included in the study were divided into two groups based on the presence of low transvalvular gradients: concordant (71.8%) and discordant (28.2%). Three patients were excluded from the discordant AS group due to moderate stenosis. The LV indices did not differ significantly between the two groups, except for ejection fraction, which was higher in the discordant AS group ($p = 0.007$) (Table 2). In the discordant AS group, the aortic root angle was more acute than that in the concordant stenosis group (114 [110; 117°] vs. 124 [118; 132°], $p < 0.001$). The Doppler intercept angle was larger in the discordant AS group (30.6 [27.5; 34.6°] vs. 18.8 [12.6; 26.0°], $p < 0.001$).

**Table 2.** Echocardiographic parameters in the concordant and discordant variants of aortic stenosis.

| Parameters | Concordant AS 56 (71.8) | Discordant AS 22 (28.2) | *p* |
|---|---|---|---|
| Left ventricle | | | |
| $EDI_{LV}$, $mL/m^2$ | 49.4 [41.0; 55.0] | 52.7 [45.9; 68.1] | 0.228 |
| $ESI_{LV}$, $mL/m^2$ | 19.2 [16.0; 29.8] | 18.9 [16.2; 23.8] | 0.787 |
| $SI_{LV}$, $mL/m^2$ | 30.6 [24.7; 35.7] | 34.1 [29.0; 43.9] | 0.082 |
| $EF_{LV}$, % | 59 [55; 64] | 65 [60; 68] | 0.007 * |
| E/A | 0.85 [0.70; 1.21] | 0.83 [0.67; 1.22] | 1.000 |
| E/e′ | 11.0 [8.5; 13.6] | 8.0 [6.6; 12.2] | 0.072 |
| Parameters of the aorta and aortic valve | | | |
| $VTI_{LVOT}$, cm | 21.7 [18.7; 25.6] | 23.2 [19.7; 25.7] | 0.702 |
| AV annulus diameter, mm | 21 [20; 23] | 22 [20; 23] | 0.788 |
| LVOT diameter, mm | 21 [20; 22] | 21 [20; 24] | 0.207 |
| Valsalva sinus diameter | 33 [31; 35] | 35 [30; 38] | 0.506 |
| Thickness of septum at basal level, mm | 18 [17; 20] | 18 [15; 20] | 0.426 |
| Aortoseptal angle, ° | 124 [118; 132] | 114 [110; 117] | <0.001 * |
| Doppler intercept angle in A5C, ° | 18.8 [12.6; 26.0] | 30.6 [27.5; 34.6] | <0.001 * |
| Aortic regurgitation, grade | 1.0 [1.0; 1.5] | 1.0 [1.0; 2.0] | 0.651 |

AS—aortic stenosis, $EDI_{LV}$—end-diastolic volume index of left ventricle, $ESI_{LV}$—end-systolic volume index of left ventricle, $SI_{LV}$—stroke index of left ventricle, $EF_{LV}$—ejection fraction of left ventricle, $VTI_{LVOT}$—the left ventricle outflow tract velocity time integral, AV—aortic valve, A5C—apical 5-chamber view. Data are presented as absolute values (*n*) and percentages (%), median (Me), and interquartile ranges (IQR). * marked significance (*p*-value < 0.05).

A comparison of velocity transvalvular indices was conducted, and in the concordant AS group, there was a coincidence of velocity and calculated indices (AVA, AVAi) obtained from both the apical window and RPW, as shown in Table 3. A statistically significant difference (*p* < 0.001) in all transvalvular parameters in the concordant and discordant AS groups was observed with the use of multiposition scanning. However, in the discordant AS group after RPW application, there was an increase in $\Delta P_{max}$ and $V_{max}$ AV in all patients, while the $\Delta P_{mean}$ increased in 95.5% of patients and VTI AV increased in 90.9% of patients. Furthermore, the AVA and AVAi indices decreased in 90.9% of patients (Figure 5).

**Table 3.** Echocardiographic parameters of transaortic flow depending on imaging window.

| Parameters | View | Concordance | | *p* |
|---|---|---|---|---|
| | | Concordant AS 56 (71.8) | Discordant AS 22 (28.2) | |
| $\Delta P_{max}$, mm Hg | A5C | 87 [76; 108] | 48 [39; 55] | <0.001 * |
| | RPW | 93 [75; 109] | 76 [68; 95] | 0.067 |
| *p* | | 0.324 | 0.324 | <0.001 * ↑ (100.0%), ↓ (0.0%) |
| $V_{max}$ AV, cm/s | A5C | 467 [428; 520] | 346 [310; 367] | <0.001 * |
| | RPW | 471 [431; 521] | 439 [412; 487] | 0.119 |
| *p* | | 0.429 | 0.429 | <0.001 * ↑ (100.0%), ↓ (0.0%) |
| $\Delta P_{mean}$, mm Hg | A5C | 53 [44; 70] | 29 [25; 33] | <0.001 * |
| | RPW | 52 [39; 63] | 43 [38; 56] | 0.175 |
| *p* | | 0.183 | 0.183 | <0.001 * ↑ (95.5%), ↓ (4.5%) |
| VTI AV, cm | A5C | 118 [103; 136] | 80 [72; 82] | <0.001 * |
| | RPW | 114 [101; 130] | 106 [97; 109] | 0.029 * |
| *p* | | 0.325 | 0.325 | <0.001 * ↑ (90.9%), ↓ (9.1%) |
| AVA (VTI), $cm^2$ | A5C | 0.61 [0.52; 0.81] | 1.19 [1.02; 1.27] | <0.001 * |
| | RPW | 0.63 [0.52; 0.82] | 0.84 [0.62; 0.92] | 0.099 |
| *p* | | 0.244 | 0.244 | <0.001 * ↑ (9.1%), ↓ (90.9%) |

**Table 3.** *Cont.*

| Parameters | View | Concordance | | *p* |
| | | Concordant AS 56 (71.8) | Discordant AS 22 (28.2) | |
|---|---|---|---|---|
| AVAi, cm$^2$/m$^2$ | A5C | 0.33 [0.27; 0.40] | 0.60 [0.58; 0.65] | <0.001 * |
| | RPW | 0.35 [0.28; 0.42] | 0.42 [0.29; 0.46] | 0.131 |
| *p* | | 0.251 | 0.251 | <0.001 * ↑ (9.1%), ↓ (90.9%) |

$\Delta P_{max}$—peak pressure gradient, A5C—apical 5-chamber view, RPW—right parasternal window, $V_{max}$ AV—peak aortic jet velocity, $\Delta P_{mean}$—mean pressure gradient, VTI AV—velocity time integral of transvalvular flow, AVA—aortic valve area, AVAi—aortic valve area index. Data are presented as absolute values (*n*) and percentages (%), median (Me), and interquartile ranges (IQR). ↑ (proportion of increase), ↓ (proportion of decrease). * marked significance (*p*-value < 0.05).

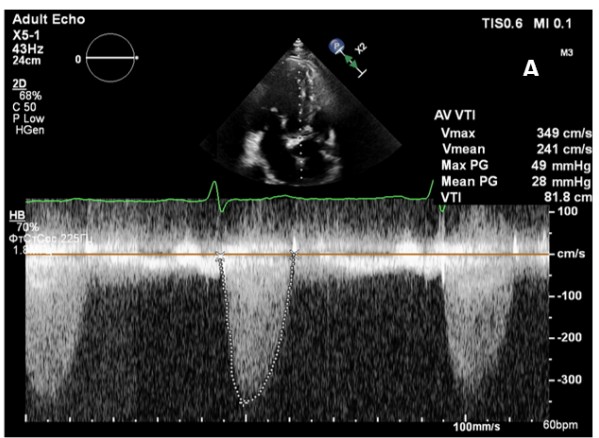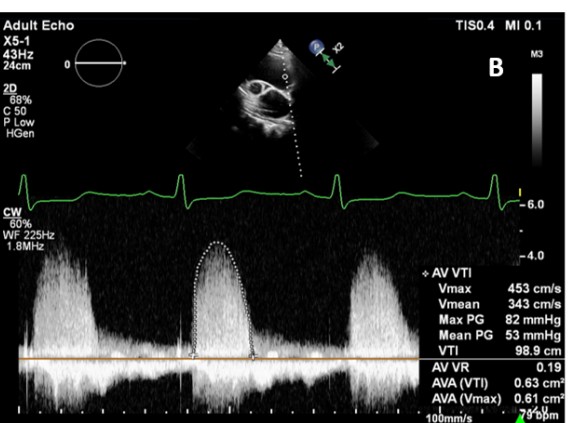

**Figure 5.** Reclassification of aortic stenosis severity in one patient. (**A**) Apical window pressure gradients corresponding to moderate stenosis, (**B**) right parasternal window pressure gradients corresponding to very severe stenosis.

When using RPW, these significant intergroup differences were mitigated and comparable to those in the concordant AS group. Specifically, low-gradient AS was reclassified to high-gradient severe or very severe AS in 22 cases (88.0%) (Figure 5). Reclassification of AS severity, including the transition from severe to critical, was observed in 30 patients (38.5% of the total cohort) (Figure 6). AS was classified as moderate stenosis in three cases (12.0%).

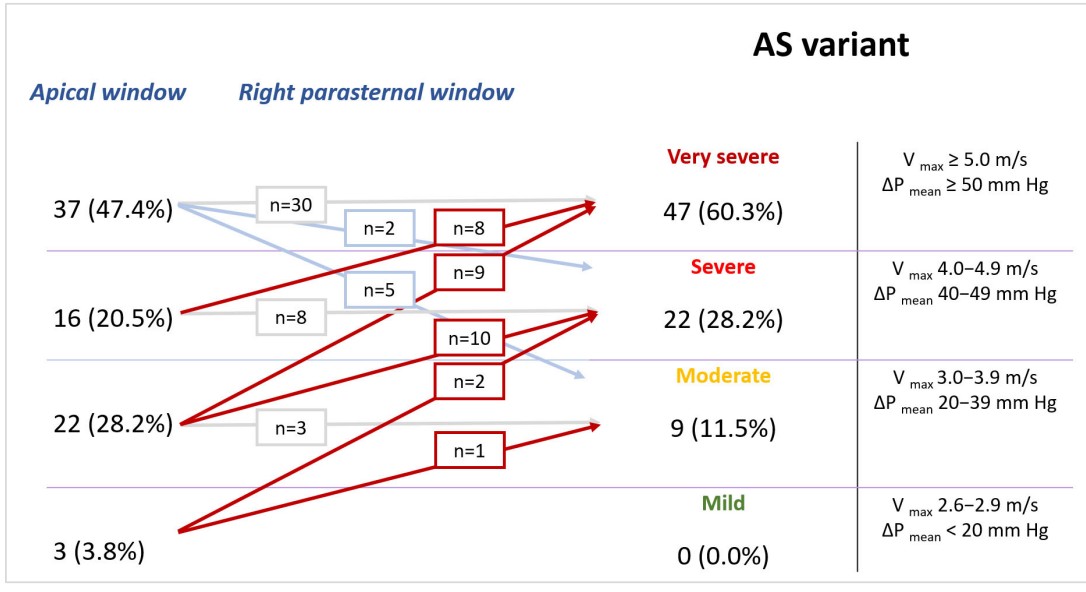

**Figure 6.** Reclassification of aortic stenosis severity depending on the imaging window.

*Reproducibility Assessment*

The variability of the test–retest data, including intra- and interobserver comparisons, is presented in Table 4.

**Table 4.** Results of intra-observer, inter-observer, and test–retest reproducibility analyses.

| Variable | ICC (95% Confidence Interval) | | |
|---|---|---|---|
| | **Intra-Observer** | **Inter-Observer** | **Test–Retest** |
| $V_{max}$ AV | 0.98 (0.99–1.0) | 0.93 (0.82–0.98) | 0.98 (0.94–0.99) |
| $\Delta P_{mean}$ AV | 0.97 (0.95–0.98) | 0.96 (0.90–0.97) | 0.97 (0.91–0.99) |
| VTI AV | 0.96 (0.9–0.99) | 0.95 (0.87–0.98) | 0.96 (0.90–0.99) |
| AV annulus | 0.99 (0.99–1.0) | 0.96 (0.89–0.98) | 0.99 (0.97–0.99) |

ICC—intraclass correlation coefficient, $V_{max}$—peak aortic jet velocity, $\Delta P_{mean}$—mean pressure gradient, VTI—velocity time integral of transvalvular flow, AV—aortic valve.

## 4. Discussion

In the presence of severe AS ($V_{max} \geq 4.0$ m/s), the rate of event-free survival over a period of 2 years is 30–50% [7]. AV replacement, either surgically or via transcatheter intervention (TAVR), is recommended for symptom management and reducing mortality in patients with severe high-gradient AS (stage D1) [8]. For asymptomatic AS, the optimal intervention timing remains controversial [9,10] and the decision to intervene requires a careful evaluation of the benefits and risks for each individual patient [3]. In the absence of adverse prognostic signs, a watchful waiting approach is usually recommended until symptoms appear [11].

Several studies, including randomized trials such as the Randomized Comparison of Early Surgery versus Conventional Treatment in Very Severe Aortic Stenosis (RECOVERY, 2020) [12] and the AVATAR study (2021) [13], have assessed the safety of a passive approach. However, these studies demonstrated clear benefits of earlier surgical intervention for asymptomatic severe AS compared to conservative treatment. Meta-analyses have also confirmed that earlier intervention reduces cardiovascular mortality and all-cause mortality compared to a watchful waiting strategy [14,15]. Prolonged pressure overload in severe AS leads to structural and functional changes in the LV, which may have unfavorable clinical consequences, such as the development of heart failure with a preserved LV ejection fraction [16]. According to Kvaslerud AB et al. (2021) [17], mortality rates of up to 10% within 1 year of follow-up and increased mid-term major adverse cardiovascular and cerebrovascular events (MACE) frequency have been reported in asymptomatic AS, raising doubts about its "benign" course.

The results confirmed the high prognostic significance of transvalvular velocity characteristics in not only assessing the probability of transitioning to the symptomatic stage of the disease, but also in stratifying the risk of adverse events [18].

The TTE analysis of the transaortic flow is a traditional method for evaluating the severity of AS, and it is considered fundamental by many researchers. However, the use of multiplane scanning is recommended for assessing the severity of AS [1]. The EACVI Scientific Committee conducted the largest analysis of visualization methods for AS. According to the results of a survey obtained from 125 centers from 32 countries [19], only half of the centers regularly used both imaging windows (apical and RPW) for velocity evaluation. This finding may require additional emphasis in future recommendations [20].

In the study by Benfari et al. (2017), which involved 330 elderly patients (mean age 81 years) with varying degrees of AS, multiposition scanning was extensively analyzed [21]. The right parasternal view was determined to be reproducible in 83% of cases. Comparing velocity measurements and AVA from the apical window and RPW, the study revealed that the apical view underestimated transaortic $V_{max}$ and $\Delta P_{mean}$ in almost 80% of patients, resulting in a larger AVA when using the continuity equation. This led to the reclassification of the severity of AS in a quarter of the patients. Furthermore, the right parasternal view identified discordant AS (low gradient) in 44% of cases, which was then reclassified as

concordant high-gradient AS. These findings suggest that multiposition scanning is an important tool for accurately assessing the severity of AS, and it should be considered when evaluating patients with this condition.

In a 2022 study, the use of the RPW in addition to the apical window to assess the severity of AS resulted in a reduced proportion of low-gradient AS. This finding suggested that relying solely on the apical method may underestimate the severity of AS [22].

Our data indicated that if only the apical window was used to assess the severity of AS, indications for surgical treatment were underestimated in 22 patients, which accounted for 28.2% of the total group. This failure to receive timely and appropriate treatment could lead to a decrease in the potential benefit of treatment and a reduction in annual event-free survival.

In cases where the aortic root has a more pronounced angulation, the flow may be distorted, hindering the proper alignment of the ultrasound beam from the apical window. Consequently, the peak velocity of the transaortic flow is more likely to be determined outside of the apical window [23].

Limited data exist on the use of a non-apical window (RPW, subcostal, suprasternal, and right supraclavicular) for evaluating the severity of AS [5,21,22,24,25]. Thaden JJ et al. conducted a study in 2015 to determine the highest peak transaortic velocity obtained from different visualization windows other than the traditional apical view [8]. In patients with greater angulation of the aortic root (<115°), $V_{max}$ was determined outside the apical window in half of the patients, with the RPW being the most frequent (65% of patients) and the apical window coming in second. The authors concluded that ignoring non-apical views could lead to incorrect classification of AS severity in 23% of cases. Similar results were obtained in Cho EJ et al.'s study in 2016 [26], which recommended adding RPW to the apical window to achieve the most accurate assessment of AS severity, particularly in patients with more pronounced aortoseptal angulation.

Accurate non-invasive assessment of peak aortic jet velocity, $\Delta P_{mean}$, and estimated AVA using Doppler echocardiography depends on proper alignment of the ultrasound beam with the direction of blood flow [27]. To obtain an accurate measurement, the Doppler transaortic flow intercept angle should ideally not exceed 20 degrees. As the angle increases, the likelihood of underestimating velocity parameters also increases. In a large study of 500 healthy subjects, the aortic septal angle was negatively correlated with age, while other anthropometric variables had no significant effect on this parameter [28]. Additionally, other aortic parameters, such as AV annulus and diameter of the ascending aorta, were determined to be related to body weight. The aortoseptal angle decreases with age, which may be part of age-related geometric changes in the thoracic aorta, including unfolding and lengthening, anterior rotation of the heart, a sigmoid-shaped interventricular septum, and interventricular septal hypertrophy. In combination, these may lead to more pronounced aortic root angulation.

Furthermore, difficulties of aortic flow detection may be due to increased calcification and deformation of the AV, as well as due to age-related emphysema, which impede visualization of the ascending aorta and the AV. Therefore, in patients with suspected low-gradient AS, regardless of the LV ejection fraction, it is important to evaluate the morphology of the AV (degree of calcification and amplitude of opening and mobility of the leaflets) and include multiple-view assessment of velocity characteristics as part of the mandatory examination protocol.

### 5. Study Limitations

This study was a prospective, single-center observational study, which limits its ability to predict the results of a randomized controlled trial. Another limitation of the study is the inclusion of patients with both a preserved LV ejection fraction and evidence of systolic dysfunction, which may have caused underestimation of pressure gradients due to decreased LV contractility and low flow. Additionally, the assumption of a circular shape

of the LVOT in calculating AVA and its indexed value may underestimate stroke volume and ultimately AVA, as the LVOT is known to be elliptical.

Despite these limitations, the RPW assessment of velocity transvalvular flow was able to provide additional information regarding the true severity of AS.

## 6. Conclusions

Accurate assessment of AS severity depends on identifying the maximal velocity characteristics on AV. However, neglecting non-apical imaging windows increases the likelihood of underestimating aortic flow characteristics and the degree of stenosis, which can result in the misclassification of AS severity. To mitigate this, the use of the RPW can effectively reduce significant discrepancies in velocity characteristics in determining the severity of AS and decrease the number of cases defined as low-gradient AS.

**Author Contributions:** Conceptualization, E.Z.G. and I.V.S.; data curation, I.V.S. and D.I.M.; formal analysis, I.V.S. and D.I.M.; investigation, I.V.S., I.Y.F. and Y.D.P.; methodology, E.Z.G. and I.V.S.; project administration, E.Z.G., I.I.S. and I.V.V.; visualization, I.V.S. and I.Y.F.; writing—original draft, I.V.S.; writing—review and editing, I.V.S., D.I.M. and D.V.M. All authors have read and agreed to the published version of the manuscript.

**Funding:** This research received no external funding.

**Institutional Review Board Statement:** The study was conducted in accordance with the Declaration of Helsinki and approved by the Ethics Committee of the A.N. Bakulev National Medical Research Center for Cardiovascular Surgery, Ministry of Health of the Russian Federation (protocol code 2 and date of approval 29 April 2022).

**Informed Consent Statement:** Not applicable.

**Data Availability Statement:** The data are not publicly available due to informed consent confidentiality paragraph.

**Conflicts of Interest:** The authors declare no conflict of interest.

## Abbreviations

| | |
|---|---|
| AS | aortic stenosis |
| $\Delta P_{mean}$ | mean pressure gradient |
| $V_{max}$ | peak aortic jet velocity |
| AVA | aortic valve area |
| AVAi | indexed aortic valve area |
| EOA | effective orifice area |
| TTE | transthoracic echocardiography |
| AV | aortic valve |
| RPW | right parasternal window |
| BSA | body surface area |
| BMI | body mass index |
| COPD | chronic obstructive pulmonary disease |
| TIA | transient ischemic attack |
| NYHA | New York Heart Association |
| IQR | interquartile range |
| LV | left ventricle |
| ASE | American Society of Echocardiography |
| EACVI | European Association of Cardiovascular Imaging |
| $CSA_{LVOT}$ | cross-sectional area of left ventricle outflow tract |
| $VTI_{LVOT}$ | the left ventricle outflow tract velocity time integral |
| $VTI_{AV}$ | velocity time integral of transvalvular flow |
| LVOT | left ventricle outflow tract |
| A5C | apical 5-chamber view |

| LA | left atrium |
|----|-------------|
| Ao | aorta |
| ICC | intraclass correlation coefficient |
| $EDI_{LV}$ | end-diastolic volume index of left ventricle |
| $ESI_{LV}$ | end-systolic volume index of left ventricle |
| $SI_{LV}$ | stroke index of left ventricle |
| $EF_{LV}$ | ejection fraction of left ventricle |
| $\Delta P_{max}$ | peak pressure gradient |
| TAVR | transcatheter aortic valve replacement |
| MACE | major adverse cardiovascular events |

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
