# Peer review of "Advantages of Multiposition Scanning in Echocardiographic Assessment of the Severity of Discordant Aortic Stenosis"

_pathophysiology, doi:10.3390/pathophysiology30020015_

Round 1

Reviewer 1 Report

The paper is focused on scanning in echocardiographic assessment of the severity of discordant aortic stenosis with the aim of comparing the severity of discordant aortic stenosis (AS) assessment using multiposition scanning and the standard apical window. The aim is coherent to the scope of the journal.

The paper is structured according to the guidelines of the journal. Even if it a small sample 104 patients who underwent preoperative transthoracic echocardiography ranked according to the degree of AS severity, I think it has been used a good methology.

I just suggest to modify table 1 in a more simple way.

All the abbreviation should be used in the text and when used for the first time they should be written in the complete form with the abbreviation in ().

Author Response

Our writing team thanks you for your positive review. As for Table 1, we would like to leave it unchanged. All the abbreviations we first used in the text, we have supplemented them with their full names. Your suggestions have been taken into account, and we thank you again for your helpful recommendations.

Reviewer 2 Report

This study is very interesting. But, the

sample size seams too small. 

Could you provide us a proof 

of sample size relevance? 

After that, the manuscript can

be accepted. 

Author Response

Thank you very much for your kind review of our study and your helpful comments.

The required number of patients was determined to compare the main parameter, ΔPmean, which is an indication for surgical treatment. As baseline values for calculating the sample size, we used data from the study of G. Benfari et al (2017), where ΔPmean was 28±14 mm Hg for A5C and 36±18 mm Hg for RPW. The present study was expected to achieve a difference in parameters no smaller than that of G. Benfari et al (2017), with a standard deviation not exceeding 18 mmHg. G*Power version 3.1.9.7, statistical test: Wilcoxon signed rank test (single sample size) was used for calculation. Power level values (1-β) ranging from 80% to 99% and a significance level of α 0.05 were considered. Under these conditions, the required sample size was 44 patients at 80% power, 58 patients at 90% power, and 100 patients at 99% power. Thus, the actual number of patients studied, which amounted to 78 patients, can be considered sufficient to solve the objectives of the study.

Faul, F., Erdfelder, E., Lang, A.-G., & Buchner, A. (2007). G*Power 3: A flexible statistical power analysis program for the social, behavioral, and biomedical sciences. Behavior Research Methods, 39, 175-191.

Benfari G, Gori AM, Rossi A, et al. Feasibility and relevance of right parasternal view for assessing severity and rate of progression of aortic valve stenosis in primary care. Int J Cardiol. 2017;240:446-451. doi:10.1016/j.ijcard.2017.04.091

Reviewer 3 Report

The paper “Advantages of multiposition scanning in echocardiographic assessment of the severity of discordant aortic stenosis” is a further confirmation of the usefulness of scanning multiple echocardiographic windows for assessing the severity of aortic stenosis. It is mandatory in the case of discordant clinical and anatomical findings.

This study does not add new data. It would be more interesting to the reader if the authors added the elliptical shape of the left ventricular outflow tract and pressure recovery phenomenon corrections.

Ringle A, Castel AL, Le Goffic C, Delelis F, Binda C, Bohbot Y, Ennezat PV, Guerbaai RA, Levy F, Vincentelli A, Graux P, Tribouilloy C, Maréchaux S. Prospective assessment of the frequency of low gradient severe aortic stenosis with preserved left ventricular ejection fraction: Critical impact of aortic flow misalignment and pressure recovery phenomenon. Arch Cardiovasc Dis. 2018 Aug-Sep;111(8-9):518-527. doi: 10.1016/j.acvd.2017.11.004.

Line 33 reproducibility feasibility is likely more correct.

Author Response

Thank you for your comment. We will take your expert opinion into account and analyze the shape of LV tract and its influence on the results obtained in the next study. We have already started to analyze different shapes of LV tract (cylindrical, funnel-shaped, or hourglass-shaped) and the effect of the chosen diameter on the results of application of the continuity equation to calculate AVA. In the present study, we measured the diameter of the LV outflow tract and placed the sample volume (pulsed-wave Doppler) strictly in the same defined location, which allowed us to minimize the influence of the overall shape of the tract on the calculation of the residual orifice area. In addition, we wanted to emphasize the importance of using nonapical imaging windows when clinical and anatomical data disagree, and we believe that in such cases, the use of additional imaging windows is mandatory.

Line 33 has been changed, corrected to "Reproducibility of feasibility", thanks for your advice.

Reviewer 4 Report

In their study, the authors discussed a method that can be used for the evaluation of gradient and valve area in aortic valve stenosis, which is a common problem in daily practice.  While the article is generally valuable , some corrections have been suggested .  It is recommended that the article be evaluated in terms of English spelling.  in which patients rpw evaluation Could be applied is there any predictable specialty in suitbale patients for rpw imaging .  in healty individuals is there any benefit of using thing to the procedure

Author Response

Thank you very much for your kind review of our study and your helpful comments.

If apical views provide adequate information in healthy subjects, there is no need to use additional imaging windows. You are very correct that estimation of gradient and valve area in aortic valve stenosis is a common problem in daily practice.  And we believe that non-apical imaging windows should be used in all questionable and discordant situations. Of all non-apical windows, we prefer RPW as the most informative in patients with disagreeing clinical and anatomical data. With RPW we were able to achieve data agreement in a significant number of patients without the use of additional costly imaging modalities.

Round 2

Reviewer 3 Report

It will be interesting to read your next paper.